# Early Detection of Hyperprogressive Disease in Non-Small Cell Lung Cancer by Monitoring of Systemic T Cell Dynamics

**DOI:** 10.3390/cancers12020344

**Published:** 2020-02-04

**Authors:** Hugo Arasanz, Miren Zuazo, Ana Bocanegra, María Gato, Maite Martínez-Aguillo, Idoia Morilla, Gonzalo Fernández, Berta Hernández, Paúl López, Nerea Alberdi, Carlos Hernández, Luisa Chocarro, Lucía Teijeira, Ruth Vera, Grazyna Kochan, David Escors

**Affiliations:** 1Oncoimmunology Group, Navarrabiomed, Instituto de Investigaciones Sanitarias de Navarra (IdISNA) UPNA, Irunlarrea st, 3, 31008 Pamplona, Spain; hugo.arasanz.esteban@navarra.es (H.A.); miren.zuazo.ibarra@navarra.es (M.Z.); ai.bocanegra.gondan@navarra.es (A.B.); gonzalo.fernandez.hinojal@navarra.es (G.F.); carlos.hernandez.saez@navarra.es (C.H.); lchocarro@alumni.unav.es (L.C.); 2Department of Medical Oncology, Complejo Hospitalario de Navarra, Instituto de Investigaciones Sanitarias de Navarra (IdISNA), Irunlarrea st, 3, 31008 Pamplona, Spain; maite.martinez.aguillo@navarra.es (M.M.-A.); Idoia.morilla.ruiz@navarra.es (I.M.); bertahernandezmarin@hotmail.com (B.H.); lucia.teijeira.sanchez@navarra.es (L.T.); 3Division of Immunology and Immunotherapy, Center for Applied Medical Research (CIMA), Instituto de Investigaciones Sanitarias de Navarra (IdISNA), University of Navarra, Pio XII ave, 55, 31008 Pamplona, Spain; mariagato13@gmail.com; 4Radiology Department, Complejo Hospitalario de Navarra, Instituto de Investigaciones Sanitarias de Navarra (IdISNA), Irunlarrea st, 3, 31008 Pamplona, Spain; p.lopez.sala@navarra.es (P.L.); n.alberdi.aldasoro@navarra.es (N.A.)

**Keywords:** hyperprogressive disease, immunotherapy, NSCLC

## Abstract

Hyperprogressive disease (HPD) is an adverse outcome of immunotherapy consisting of an acceleration of tumor growth associated with prompt clinical deterioration. The definitions based on radiological evaluation present important technical limitations. No biomarkers have been identified yet. In this study, 70 metastatic NSCLC patients treated with anti-PD-1/PD-L1 immunotherapy after progression to platinum-based therapy were prospectively studied. Samples from peripheral blood were obtained before the first (baseline) and second cycles of treatment. Peripheral blood mononuclear cells (PBMCs) were isolated and differentiation stages of CD4 lymphocytes quantified by flow cytometry and correlated with HPD as identified with radiological criteria. A strong expansion of highly differentiated CD28**^−^** CD4 T lymphocytes (CD4 THD) between the first and second cycle of therapy was observed in HPD patients. After normalizing, the proportion of posttreatment/pretreatment CD4 THD was significantly higher in HPD when compared with the rest of patients (median 1.525 vs. 0.990; *p* = 0.0007), and also when stratifying by HPD, non-HPD progressors, and responders (1.525, 1.000 and 0.9700 respectively; *p* = 0.0025). A cut-off value of 1.3 identified HPD with 82% specificity and 70% sensitivity. An increase of CD28**^−^** CD4 T lymphocytes ≥ 1.3 (CD4 THD burst) was significantly associated with HPD (*p* = 0.008). The tumor growth ratio (TGR) was significantly higher in patients with expansion of CD4 THD burst compared to the rest of patients (median 2.67 vs. 0.86, *p* = 0.0049), and also when considering only progressors (median 2.67 vs. 1.03, *p* = 0.0126). A strong expansion of CD28**^−^** CD4 lymphocytes in peripheral blood within the first cycle of therapy is an early differential feature of HPD in NSCLC treated with immune-checkpoint inhibitors. The monitoring of T cell dynamics allows the early detection of this adverse outcome in clinical practice and complements radiological evaluation.

## 1. Introduction

Immune checkpoint inhibitors (ICIs) have revolutionized the treatment of cancer, as they have been proven to be more efficacious and less toxic than conventional chemotherapy or targeted therapies in several tumor types. However, some patients experience an unexpected accelerated tumor growth during treatment associated with prompt clinical deterioration. This phenomenon has been defined as hyperprogressive disease (HPD) [1,2,3]. The underlying mechanisms are still not clearly defined, although some recent studies are shedding light upon the matter. Tumor infiltration by M2-like CD163^+^ CD33**^+^** PD-L1**^+^** macrophages was found in tumor biopsies from all non small cell lung cancer (NSCLC) patients exhibiting hyperprogression. Tumor progression was shown to correlate with anti-PD1 Fc interaction with Fc receptors in macrophages in a murine xenograft model [4]. A recent translational study in gastric cancer found an increase of proliferating Ki67**^+^** regulatory T cells (Tregs) in tumor samples of HPD patients, and confirmed stronger immune suppressive activities of these cells in vitro after anti-PD1 exposure [5].

Several definitions for HPD based on radiological criteria, sometimes complemented with clinical outcomes, have been proposed so far [1,2,6,7]. Most of them quantify the change in tumor growth rates before and after immunotherapy, and consider HPD when a predetermined cut off value of increment in tumor growth rate is surpassed. However, due to the very nature of radiological evaluations, all criteria face important limitations that preclude their applicability in different situations. A progression caused by non-measurable disease or the unavailability of previous radiological tests are amongst different drawbacks. Additional biological information that could be obtained using noninvasive techniques includes the evaluation of the immunological *status quo*. This information might complement the definition of HPD and help to improve and even anticipate its detection.

We have recently demonstrated that functional systemic CD4 T cell immunity is required for a clinical response to immune checkpoint inhibitors in NSCLC patients who had progressed to platinum-based chemotherapy [8]. Hence, we classified patients in two groups according to systemic CD4 T cell functionality. G1 patients showed proficient CD4 T cell immunity before the start of immunotherapy, characterized by good proliferative responses and low coexpression of PD-1 and LAG-3 after activation. On the other hand, G2 patients showed highly dysfunctional CD4 responses, with strong coexpression of PD-1 and LAG-3. While G2 patients did not respond to immunotherapy, G1 patients included objective responders who showed an expansion of the CD28+ T cell populations after the first cycle of therapy [8], in agreement with other studies [9,10]. The expression of CD28 in T cells is frequently used as a maker of differentiation. Thus, poorly- and intermediately-differentiated T cells express CD28, while highly differentiated cells do not [11,12]. Indeed, G1 and G2 patients can be readily identified by quantifying the relative percentage of CD4 T_HD_ cells within CD4 cells in peripheral blood before starting immunotherapies. While G1 patients are characterized by having more than 40% of CD4 T_HD_, G2 patients show less than 40% CD4 T_HD_ [8]. This classification constituted a first criterion for the identification of non-responder versus potential responder patients. We also found that all patients presented systemic CD4 T cells with mainly Th17 phenotypes, without differences between responders and progressors, and without significant differences in the percentage of lung-cancer-specific CD4 or CD8 T cells [8]. However, CD4 T cells from progressors were highly dysfunctional in proliferation but not in cytokine production. 

While an expansion of CD28+ populations marks responders [8,9,10], it is likely that an expansion of CD28- populations (highly-differentiated T cells) is a marker for failure regarding immunotherapy. The present study furthers analyzes the immune profiles of our cohort for diagnostic markers of HPD.

## 2. Results

### 2.1. Patients and Clinical Outcomes

In the present study, a cohort of 70 lung cancer patients were included, i.e., 51 (72.9%) with nonsquamous NSCLC and 19 (27.1%) with squamous NSCLC. Atezolizumab was the most frequently prescribed treatment (33 patients; 47.1%), followed by nivolumab (28 patients; 40%), and pembrolizumab (9 patients; 12.9%). The majority (50 patients; 71.4%) had received one previous line of treatment for metastatic disease. Fifteen patients (21.4%) were of poor prognosis according to the Gustave Roussy Immune Score (GRIm-Score) [13], while 37 (52.9%) had good prognostic characteristics. According to the Lung Immune Prognostic Index (LIPI), 5 patients (7.1%) were of poor prognosis while 57 (81.5%) were classified as good or intermediate prognosis. According to G1/G2 classification on systemic CD4 T cell functionality as defined by Zuazo M et al. [8], 38 patients (52.9%) presented a G1 profile (>40% baseline CD4 T_HD_) and 31 (44.3%) had a G2 profile (<40% baseline CD4 T_HD_) (Table 1).

Overall response rate (ORR) was 24%, and 9% of the patients presented stable disease (SD). Median progression-free survival (mPFS) was 8.9 weeks, and median overall survival (mOS) was 48.1 weeks.

The incidence of HPD in our cohort study was 17.9% (95% CI 9.6 to 29.4). In HPD patients, the mPFS was significantly lower (6 weeks; 95% CI 4.9 to 7.1) compared with all other patients (10.9 weeks; 95% CI 6.3 to 15.4) (*p* < 0.001) (Figure 1A), even when considering only progressors (*p* = 0.044) (Figure 1B). mOS in HPD patients was 14.0 weeks (95% CI, 6.5 to 21.5) and 54.7 weeks (95% CI, 36.7 to 72.8) for all other patients (*p* = 0.006) (Figure 1C).

The association of HPD with G1/G2 CD4 T cell profiles was analyzed in the whole cohort of patients (Figure 2A), as well as with other variables (Appendix A). While objective responders were found in G1 patients (>40% baseline CD4 T_HD_), HPD was detected within G2 patients. No significant association was observed with baseline CD8 T_HD_ profiles (Figure 2B). Indeed, HPD was very significantly associated with a baseline G2-type of systemic CD4 T cell profile as defined by Zuazo et al. (*p* = 0.003 by Pearson’s Chi Square test) within progressors, which identifies patients with dysfunctional CD4 immunity before the start of immunotherapy [8]. The HPD proportion was of 3.3% for patients with G1 profile and 37.5% for patients with G2 profile (Figure 1C). A significant correlation was found between HPD and smoking (*p* = 0.035). No interaction was found between HPD and the immunotherapy drug (*p* = 0.440), GRIm (*p* = 1) or any of its variables, i.e., LIPI of poor prognosis (p = 1.000), number of previous treatments (p = 0.151), gender (*p* = 1.000), age (*p* = 0.072), performance status (*p* = 0.189), tumor histology (*p* = 1.000), PD-L1 tumor expression (*p* = 0.599), number of affected organs (*p* = 0.707) or liver metastases (*p* = 0.707).

### 2.2. Systemic Expansion of CD28-Negative CD4 T Cells (T_HD_ Cells) within the First Cycle of Therapy Is Significantly Associated with HPD

Although a significant correlation of HPD with a baseline G2 profile was observed (Figure 2), it was not sufficient to separate G2 progressors from hyperprogressors. Along with others, we have shown that a systemic expansion of CD28+ CD4 T cells following the start of immunotherapy is characteristic in responder patients. Therefore, we decided to determine whether changes in the highly differentiated CD28-negative (T_HD_) CD4 populations from baseline to first cycle of therapy would be an indicator to further discriminate HPD from non-HPD progressors.

We observed that 13 (46.4%) of the patients with a G2 profile presented a sharp expansion of the highly differentiated CD4 T cell compartment which correlated with very poor outcome (Figure 3A,B). After normalizing the data, we then compared the relative change of the percentage in CD4 T_HD_, as defined in materials and methods. When stratified by non-HPD and HPD using radiological criteria, ΔCD4 T_HD_ was significantly higher for the latter (median 1.525 vs. 0.990; *p* = 0.0007) (Figure 3C). Interestingly, HPD patients also presented a higher ΔCD4 T_HD_ when comparing with responders and with non-HPD progressors (median 1.525, 0.970 and 1.000 respectively; *p* = 0.0025), while no differences were found between these two last groups of patients (*p* > 0.05 by Bonferroni’s Multiple Comparison Test) (Figure 3D). ROC analysis provided a cut-off value of ΔCD4 T_HD_ ≥ 1.3 to identify HPD with 82% specificity and 70% sensitivity, with an area under the curve (AUC) of 0.792 (Figure 3E). Accordingly, an increase of ΔCD4 T_HD_ ≥ 1.3 (from now “CD4 T_HD_ burst”) was very significantly associated with HPD (*p* = 0.008 by Fisher’s exact test) (Figure 3F).

### 2.3. Association between CD4 T_HD_ Cell Expansion and Clinical Outcomes

To confirm the role of CD4 T cell dynamics in the response to immunotherapy in NSCLC, we evaluated the correlation of CD4 T_HD_ burst with clinical outcomes. Patients presenting a CD4 T_HD_ burst following the first cycle of PD-L1/PD-1 blockade had significantly shorter mPFS compared to the rest of patients (6.29 vs. 9.86 weeks, *p* = 0.001). These patients had all progressed within 11 weeks after the first cycle of treatment (Figure 3E). No significant differences in PFS were found when considering only progressors (*p* = 0.250), although the curves were similar to those obtained by radiological identification of HPD by TGR (Figure 4A). Likewise, a trend towards decreased mOS was observed in the cohort of patients exhibiting a CD4 T_HD_ burst (56.1 weeks; 95% CI 30.2 to 82.1 vs. 21.9 weeks; 95% CI 0 to 59.8. *p* = 0.095) (Figure 4B).

From all the variables evaluated in our study, the burst of CD4 T_HD_ cells was very significantly associated with patients with G2 profile (46.4% vs. 0%, *p* < 0.0001) and with patients treated with atezolizumab than with pembrolizumab or nivolumab (41.7% vs. 11.1% vs. 6.7% respectively, *p* = 0.005), and significantly associated with PD-L1 tumor expression < 5% (34.6% vs. 9.1%, *p* = 0.036). The hazard ratio for progression or death of patients with CD4 T_HD_ burst maintained statistical significance by multivariate analyses (HR 6.749, 95% CI 1.678 to 27.139; *p* = 0.007), when adjusted for gender, smoking habit, tumor histology, immunotherapy drug, line of treatment, number of organs affected, liver metastases, NLR, serum LDH, serum albumin, GRIm Score, G2 lymphocyte profile, and radiological HPD, as did PD-L1 expression ≥ 5%, with HR of 0.287 (95% CI 0.093 to 0.887).

To confirm that the CD4 T_HD_ burst was associated with a greater increase in tumor burden, we compared the TGRs in non-responder patients with and without CD4 T_HD_ burst. As expected, the TGR was significantly higher for patients with observed CD4 T_HD_ burst (1.70 vs. 0.86, *p* = 0.0119) (Figure 4C). This difference was similar and kept statistical significance when only progressors were included (median 1.70 vs. 1.03, *p* = 0.0126) (Figure 4C). Finally, we analyzed the value of CD28-negative CD4 cell quantification as a complementing factor for the diagnosis of HPD in combination with radiological criteria. Patients who presented both HPD by TGR and CD4 T_HD_ burst had significantly lower OS when compared with the rest of patients (24-week OS 15.2% vs. 76.1%, *p* = 0.016) (Figure 4D) and median PFS when considering only progressors (8.1 vs. 6 weeks, *p* = 0.033) (Figure 4E).

These findings support the importance of establishing an immunological profile based on quantification of highly-differentiated CD4 populations. The dynamic changes of highly differentiated CD4 T cell subpopulations induced by immunotherapy are associated with adverse outcomes to PD-1/PD-L1 targeted immunotherapy, with a good correlation with radiological criteria.

## 3. Discussion

Immunotherapy is fast advancing to front-line therapy in NSCLC. Two studies, including our previous one, showed that a systemic expansion of CD28+ T cell populations occurs in objective responders to PD-L1/PD-1 blockade therapies [8,14]. Therefore, in this exploratory study, we decided to monitor whether an increase in CD28-negative populations was a concomitant marker for failure of PD-L1/PD-1 immunotherapy, and more specifically, to hyperprogressive disease. HPD is one of the most serious adverse events of high clinical importance, but is the subject of intense debate. An incidence of HPD in NSCLC between 10–25% has been previously reported [4,7,15,16], which is in agreement with our observations. Different radiological criteria have identified HPD and its association with markedly worse PFS and OS [1,2,6,7,15,16]. However, the intrinsic characteristics of radiological evaluation, along with the limitations of RECIST criteria, make the diagnosis of HPD challenging, with dramatic consequences. The patient must have at least one measurable lesion in the basal CT scan. To detect the increase in growth of target lesions, a previous radiological control is required, which is not always available. Importantly, tumor growth during this pre-immunotherapy period might have been slowed down by the treatment received, and might be a confounding factor that causes an overestimation of HPD incidence. New lesions that appear after the beginning of immunotherapy are not considered, nor are non-measurable lesions that might significantly increase tumor burden. As a consequence, some HPD patients will not be detected in routine clinical practice. Finally, HPD might be underestimated when tumor burden is high, and the opposite if tumor burden is low, as small absolute changes in diameter would entail much greater relative variations.

Several recent preclinical and clinical studies have put forward immunological mechanisms underlying HPD of varying nature, thereby giving strength to previously published works on this matter [4,5]. The identification of these mechanisms is of the great value. However, these studies rely on biopsies, and it is difficult to translate their data into clinical practice. The identification of systemic immunological biomarkers that would help in identifying HPD would also enable an earlier diagnosis that could complement radiological evaluations.

In agreement with other studies [14], we have recently shown that CD28+ CD4 T cell populations expand in peripheral blood in objective responders to PD-L1/PD-1 blockade therapies [8]. Systemic CD4 T cell profiling allowed us to select groups of patients who were susceptible for PD-L1/PD-1 blockade. Therefore, here, we tested whether the concomitant expansion of CD28-negative CD4 T cells was a biomarker for progression. We found a good association of expansion of this population in peripheral blood with hyperprogressive disease. More specifically, a significant increase in CD28-negative CD4 T cells between baseline and the first cycle of therapy (CD4 T_HD_ burst) was associated with HPD. In this study, we also quantified the relative proportions and changes of other systemic immune populations including neutrophils, monocytes, HLA DR+ cells, and absolute T cell counts. No significant associations were found, as corroborated also by a lack of association with neutrophil-to-lymphocyte ratio, a typical prognostic variable.

Besides its correlation with HPD, we found that all patients with a systemic CD4 T_HD_ burst present disease progression within two months. In fact, the CD4 T_HD_ burst was an independent variable associated with progression by multivariate analysis. Nevertheless, our prospective study is exploratory in nature with a limited number of HPD patients. In addition, the identification of HPD is challenging, and not all studies follow the same criteria. Indeed, many HPD patients progress so rapidly that they cannot be objectively diagnosed before death. Therefore, if validated, the quantification of this systemic immunological phenomenon would lead to the interruption of treatment. To our knowledge, this is the first study associating HPD and the systemic dynamics of CD4 T cell populations. Furthermore, no prospective works regarding this issue have previously been published. Recently, Kim and collaborators linked HPD with the expansion of severely exhausted PD-1+ TIGIT+ CD8+ T cells in Asian populations [15,16]. The different immunological characteristics that have been associated with HPD in several studies indicate that it is a complex phenomenon with different players in the immune system. Indeed, all of these immunological phenomena may be different aspects of an otherwise a common dysfunctional mechanism of immunotherapies leading to HPD, as T cell dysfunction seems to be a shared consequence of all the proposed mechanisms. Although our study does not address whether CD4 T_HD_ cells are directly involved in the onset of HPD, these cells provide a biomarker that changes within the G2 cohort in HPD patients.

## 4. Materials and Methods

### 4.1. Study Design

The study was approved by the Ethics Committee at the Hospital Complex of Navarre (Pyto2017/49). Informed consent was obtained from all subjects, and all experiments conformed to the principles set out in the WMA Declaration of Helsinki and the Department of Health and Human Services Belmont Report. Samples were collected by the Blood and Tissue Bank of Navarre, Health Department of Navarre, Spain. Data and blood samples were prospectively collected from 70 Caucasian patients with locally advanced or metastatic NSCLC treated with ICI (nivolumab, pembrolizumab, atezolizumab) following current indications [17,18,19] at the Complejo Hospitalario de Navarra between September 2017 and May 2019.

Eligible patients were 18 years of age or older, and had all progressed to first-line platinum-based chemotherapy or concurrent chemoradiotherapy. A basal CT scan before the beginning of immunotherapy and a previous one were analyzed. Exclusion criteria were previous immunotherapy treatment or the existence of synchronous neoplasms.

Four milliliters of peripheral blood samples were obtained immediately prior to the infusion of the first cycle of immunotherapy and before the administration of the second. Follow-up concluded with the withdrawal of consent or after death of the patient.

### 4.2. Sample Processing and Flow Cytometry

Surface and intracellular flow cytometry analyses were performed as described [20,21]. Four milliliters of blood samples were collected from each patient, and PBMCs were isolated by FICOL gradients right after the blood extraction. PBMCs were washed and cells immediately stained with the indicated antibodies in a final volume of 50 μL for 10 min in ice. Cells were washed twice, resuspended in 100 μL of PBS, and analyzed immediately.

The following fluorochrome-conjugated antibodies were used at 1:50 dilutions unless otherwise stated: CD4-APC-Vio770 (clone M-T466, reference 130-100-455, Miltenyi Biotec, Bergisch Gladbach, Germany), CD3-APC (clone REA613, reference 130-113-135, Miltenyi Biotec), CD27-PE (clone M-T271, reference 130-093-185, Miltenyi Biotec), CD28-PECy7 (clone CD28.2, reference 302926, BioLegend, San Diego, CA, USA), CD8-FITC (clone SDK1, reference 344703, BioLegend).

Relative percentage of T cell differentiation subsets based on expression of these markers were quantified using FlowJo [11,12]. The changes in the relative T cell differentiation subsets after the first cycle of immunotherapy were normalized and expressed as a proportion of the percentage of the T cell subset posttreatment divided by the pretreatment percentage (% CD4_HD_ posttreatment / % CD4_HD_ pretreatment), hereafter referred to as ΔCD4 T_HD_.

### 4.3. Radiological Evaluation

Assessment of response was performed following standard protocols, and according to common clinical practice as decided by the treating oncologist. Tumor responses were evaluated according to RECIST 1.1 [22] and Immune-Related Response Criteria [23]. The TGR was calculated according to the definition by Ferté et al. [24], and measured before and after ICI treatment. HPD was defined as Immune-Related Response Criteria progression at first radiological evaluation with a ≥2-fold increase of the TGR after the beginning of immunotherapy, as proposed by Champiat S et al. [1].

### 4.4. Statistical Analyses

The association between qualitative variables was evaluated with Pearson’s Chi Squared test. The correlation of quantitative variables was studied with a t test for independent samples/ANOVA (if normally distributed) or *U* of Mann–Whitney/Kruskal–Wallis (if not normally distributed, or data with intrinsic high variability). Two-tailed tests were applied. Survival curves were represented by Kaplan-Meier plots, and compared between cohorts with log-rank tests. Hazard ratios were estimated by Cox regression models. Receiver operating characteristics (ROC) analysis was performed with the proportion that represented the changes in T cell subsets after ICI treatment.

## 5. Conclusions

Our results support HPD as a specific deleterious outcome of PD-1/PD-L1 targeted immunotherapy, and imply the participation of CD4 T cells. We also confirm that the radiological criterion of HPD by TGR calculation accurately predicted fast progression and correlated with lower OS in our cohort. A CD4 T_HD_ burst is associated with HPD, and its combination with radiological criteria might advance procedures used in the detection of HPD and facilitate decision making in controversial situations.

## Figures and Tables

**Figure 1 cancers-12-00344-f001:**
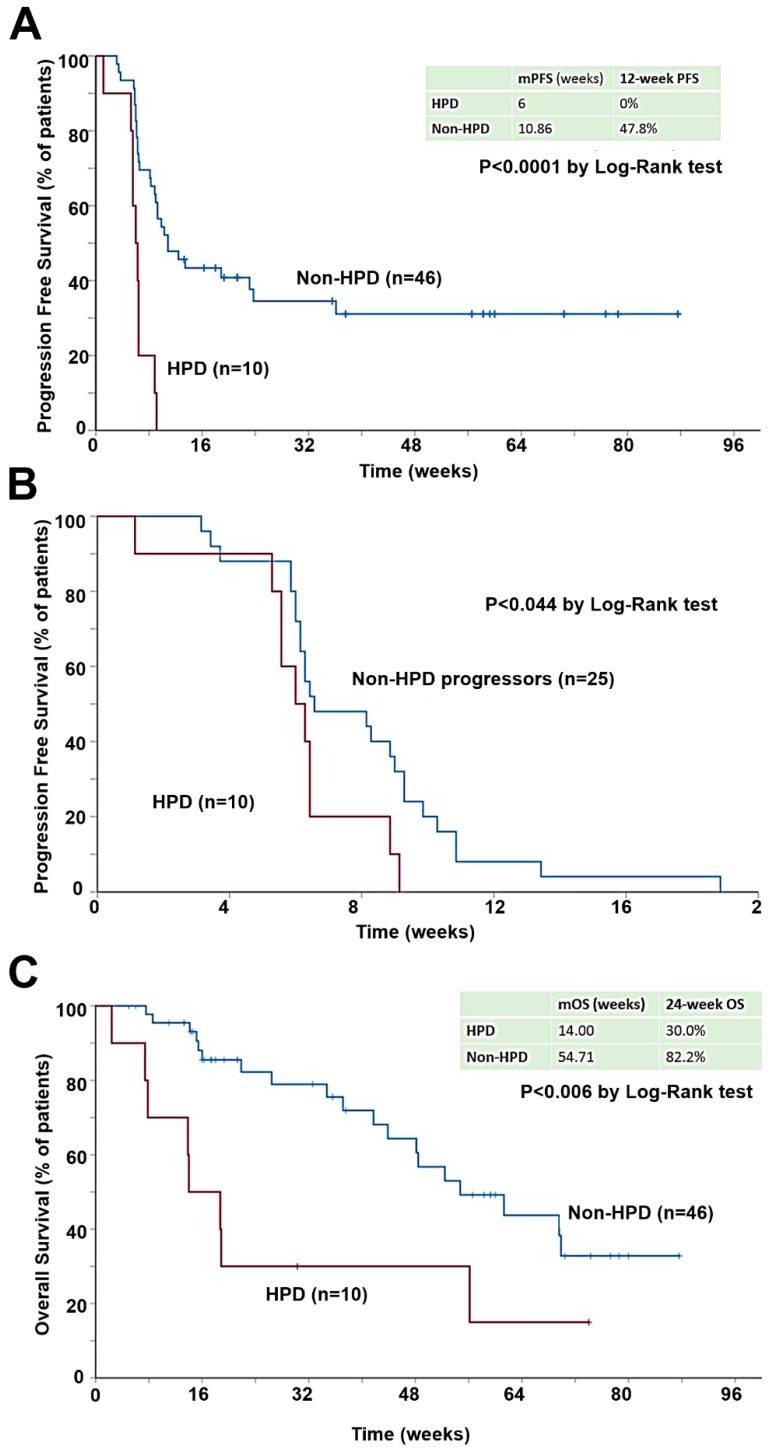
Hyperprogressive disease as defined by TGR associates with worse PFS and OS. (**A**) Kaplan-Meier plot for PFS in patients with measurable disease by RECIST 1.1 treated with immunotherapy, stratified by HPD at first radiological evaluation. (**B**) Kaplan-Meier plot for PFS only representing those patients with measurable disease by RECIST 1.1 who did not respond to immunotherapy. (**C**) Kaplan-Meier plot for OS in patients with measurable disease by RECIST 1.1 treated with immunotherapy, stratified by HPD at first radiological evaluation.

**Figure 2 cancers-12-00344-f002:**
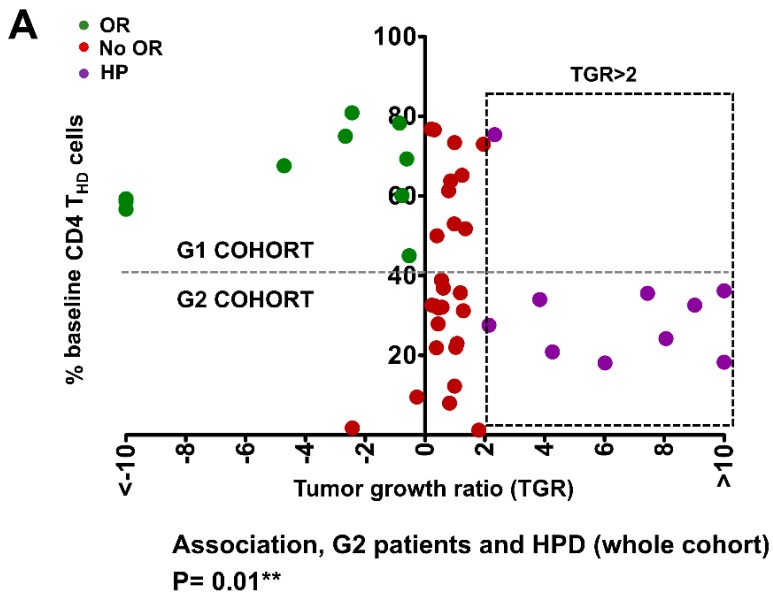
Hyperprogressive disease as defined by TGR associates with G2 baseline CD4 T_HD_ profiles. (**A**) Dot plot of color-coded clinical outcomes of the patients in our cohort represented by their baseline percentage of CD4 T_HD_ in peripheral blood and TGR. The square with dotted lines represents patients with TGRs higher than 2, the most commonly used cut-off value to separate progressors from hyperprogressors. The horizontal dotted line separates the G1 cohort (>40% CD4 T_HD_) from the G2 cohort (<40% CD4 T_HD_). The association of HPD with G2 profiles including also responders is shown below by the Fisher’s exact test. OR, objective responses; No OR, no objective responses; HP, hyperprogressors. (**B**) As in (**A**) but plotting the baseline percentage of CD8 T_HD_ cells. No significant association with HPD is observed. (**C**) Contingency table representing the incidence of HPD in patients with measurable disease by RECIST 1.1, according to G1 or G2 lymphocyte profile as defined by Zuazo M. The numbers indicate the following: absolute number of patients, row percentage, and 95% confidence interval of each percentage.

**Figure 3 cancers-12-00344-f003:**
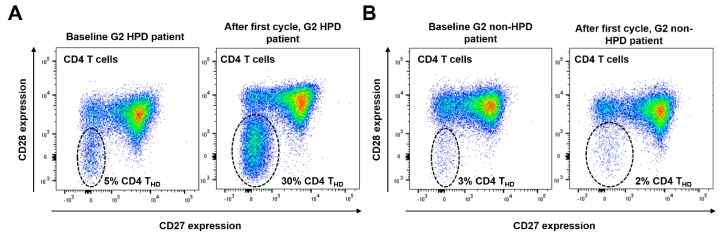
An expansion of the CD28^negative^ CD4 T cell compartment following the first cycle of immunotherapy correlates with radiological HPD. (**A**) Example of quantification of CD4 T cells according to CD28 and CD27 expression profiles in a HPD patient before starting immunotherapy (left plot, baseline) and after the first cycle of treatment (right plot); CD4 T_HD_ cells are encircled and correspond to doubly CD28/CD27-negative cells. An increase in CD4 T_HD_ cells was observed (from 5% to 30%, as indicated). (**B**) Same as (**A**) but from a G2 non-HPD progressor. No increase in CD4 T_HD_ cells was observed. (**C**) Bar graph representing the mean and 95% confidence interval of the CD4 T_HD_ proportion in patients who presented HPD compared with non-HPD progressors. (**D**) Bar graph representing mean and 95% confidence interval of the proportion of basal and post-first cycle CD4 T_HD_ cells in patients with partial response, stable disease, progressive disease or HPD. (**E**) ROC analysis of CD4 T_HD_ proportion as a function of radiological HPD. (**F**) Kaplan-Meier plot for PFS in patients with lymphocyte subpopulations quantified before and after the first cycle of immunotherapy, stratified by the detection of CD4 T_HD_ burst (≥1.3). PR: partial response. SD: Stable disease. PD: Progressive disease. HPD: Hyperprogressive disease. **; ***, indicate very significant (*p* < 0.01) and highly significant (*p* < 0.001) statistical differences respectively.

**Figure 4 cancers-12-00344-f004:**
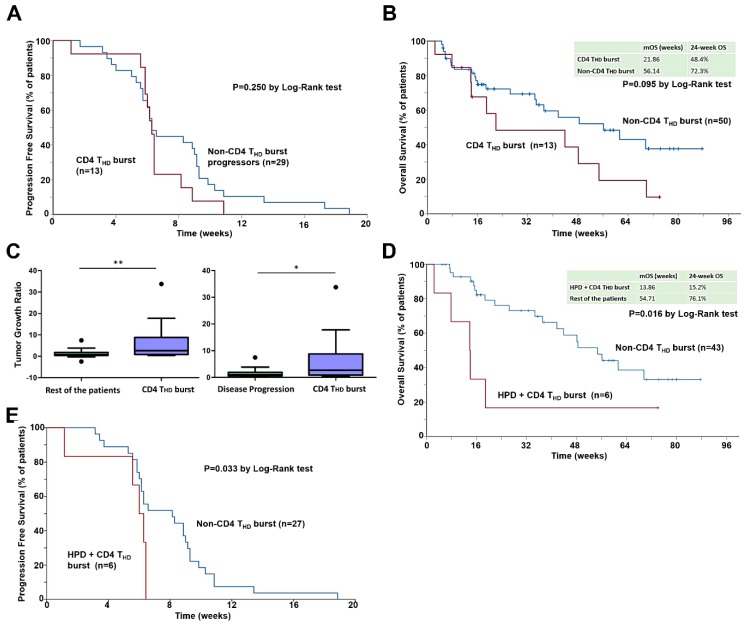
The CD4 T_HD_ burst is associated with HPD, and complements radiological criteria for its diagnosis. (**A**) Kaplan-Meier plot for PFS only representing patients with processed lymphocyte subpopulations quantified before and after the first cycle of immunotherapy who did not respond to immunotherapy, stratified by the incidence of CD4 T_HD_ burst (≥1.3). (**B**) Kaplan-Meier plot for OS in patients with lymphocyte subpopulations quantified before and after the first cycle of immunotherapy, stratified by detection of CD4 T_HD_ burst (≥1.3). (**C**) Left graph, box and whiskers plot (Tukey) representing median TGR of patients with CD4 T_HD_ burst compared to the rest of patients. Box depicts interquartile range, whiskers add up 1.5 interquartile range. Outliers are represented by dots. Right graph, same as left but comparing median TGR of patients with CD4 T_HD_ burst compared to the patients that presented progressive disease as best response. (**D**) Kaplan-Meier plot for OS in patients with measurable disease by RECIST 1.1 and lymphocyte subpopulations quantified before and after the first cycle of immunotherapy, stratified by the presence of both HPD by TGR and the detection of CD4 T_HD_ burst (≥1.3). (**E**) Kaplan-Meier plot for PFS representing patients with measurable disease by RECIST 1.1 and lymphocyte subpopulations quantified before and after the first cycle of immunotherapy, who presented progressive disease as best response, stratified by the presence of both HPD by TGR and the detection of CD4 T_HD_ burst (≥1.3). *, **, indicate significant (*p* < 0.05) and very significant (*p* < 0.01) statistical differences.

**Table 1 cancers-12-00344-t001:** Main characteristics of the cohort.

Variable	Number of Patients (%)
Age	
31–40	1 (1.4%)
41–50	2 (2.9%)
51–60	15 (21.4%)
61–70	37 (52.9%)
71–80	13 (18.6%)
>80	2 (2.9%)
Gender	
Female	18 (25.7%)
Male	52 (74.3%)
ECOG Performance status	
0–1	56 (80%)
2–4	14 (20%)
Smoking Habit	
No	6 (8.6%)
Yes	64 (91.4%)
Tumor Histology	
Non-squamous	51 (72.9%)
Squamous	19 (27.1%)
Stage	
Locally Advanced	1 (1.4%)
Metastatic	69 (98.6%)
Mutational Status	
No	68 (97.1%)
EGFR mutated	1 (1.4%)
ROS1 translocated	1 (1.4%)
Tumor PD-L1 expression	
0%	23 (32.9%)
1–4%	7 (10%)
5–49%	13 (18.6%)
≥50%	11 (15.7%)
Non-evaluable	16 (22.9%)
Immunotherapy Drug	
Atezolizumab	28 (40%)
Nivolumab	33 (47.1%)
Pembrolizumab	9 (12.9%)
Treatment Line	
2nd	50 (71.4%)
3rd	16 (22.9%)
4th or further	4 (5.7%)
Number of Tumor Sites	
≤2	21 (30%)
≥3	49 (70%)
Liver Metastases	
No	51 (72.9%)
Yes	19 (27.1%)
Neutrophil to Lymphocyte Ratio (NLR)	
≤6	51 (72.9%)
>6	19 (27.1%)
Lactate Dehydrogenase (serum)	
≤ULN	14 (20%)
>ULN	24 (34.3%)
Non Available	32 (45.7%)
Albumin (serum)	
≥ 3.5 g/dl	41 (58.6%)
< 3.5 g/dl	12 (17.1%)
Non Available	17 (24.3%)
GRIm Score	
0–1	37 (52.9%)
2–3	15 (21.4%)
Non Evaluable	18 (25.7%)
Derived Neutrophil to Lymphocyte Ratio (dNLR)	
<3	54 (77.1%)
>3	16 (22.9%)
Lung Immune Prognostic Index (LIPI)	
Intermediate/Good Prognosis	57 (81.4%)
Poor prognosis	5 (7.1%)
Non evaluable	8 (11.4%)
Lymphocytes	
Non-lymphopenia	48 (68.6%)
Lymphopenia	22 (31.4%)
Lymphocyte Profile	
G1	37 (52.9%)
G2	31 (44.3%)
Non-evaluable	2 (2.9%)

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
