# Peer review of "Early Detection of Hyperprogressive Disease in Non-Small Cell Lung Cancer by Monitoring of Systemic T Cell Dynamics"

_cancers, 2020, doi:10.3390/cancers12020344_

Round 1
Reviewer 1 Report
This manuscript entitled "Early diagnosis of hyperprogressive disease in non-small cell lung cancer by monitoring of systemic T cell dynamics" suggested the expansion of highly differentiated CD28- CD4 T helper cells in blood after first-cycle immunotherapy could be considered as a diagnosis marker for the hyperprogressive disease of the NSCLC patients receiving immunotherapy. This is an interesting prospective clinical study and may contribute to the decision making for the NSCLC patients with metastasis and recurrence after first-line therapy. Several questions and suggestion were listed as following.
The blood samples were obtained before and after immunotherapy and the THD burst in blood was observed after immunotherapy. The THD burst looks more like a prognostic marker, but not a diagnostic marker for HPD. The patient number in distinct comparisons are different. The authors should explain the criteria for the included patients in each statistic comparison. In Figure 2A, the figure legend is confusing. The left panels of flow cytometry data showed an increase of CD28-/CD27- cells population in the HPD patient, but not CD28+/CD27+ cell population. Figure 2A was not cited in the text. In Figure 2C, the lines indicating the comparisons between groups are confusing. If this study was approved by an Institutional Review Board (IRB), the approval number should be mentioned in the manuscript.Author Response
RESPONSE TO REVIEWER 1
We appreciate the overall positive comments from Reviewer 1, who considers the study to be an interesting prospective clinical study that may contribute to decision making for NSCLC patients. Reviewer 1 asks some questions and suggestions to be addressed as follows:
Reviewer one points out that Thd burst is more a prognostic maker but not a diagnostic marker for HPD.
We agree with the Reviewer, it could be considered a prognostic marker that can help for clinical decision-making. To clarify this issue, we have modified the manuscript as follows:
* The title has been changed to “Early detection of hyperprogressive disease….”.
* In the abstract, “No diagnostic biomarkers have been identified yet” has been changed to “no biomarkers have been identified yet.”
Apart from this, the rest of the manuscript does not have direct references to this marker being diagnostic or predictive.
The reviewer points out that patient number in distinct comparisons are different, and that the criteria to include patients should be made apparent in each statistics comparison.
We agree with the reviewer that it should be made apparent. The reason for this was that some patients had non-measurable disease and could not be evaluated. Therefore, we have introduced brief explanations in the figure legends.
The reviewer points out that the figure legend corresponding to Figure 2A is confusing, as well as the flow cytometry data. The left panels of flow cytometry data showed an increase of CD28-/CD27- cells population in the HPD patient, but not CD28+/CD27+ cell population. Figure 2A was not cited in the text. In Figure 2C, the lines indicating the comparisons between groups are confusing.
We sincerely apologise to the reviewer. This was caused by some errors in the figures and the assumption that readers of this paper would had read our previous paper (Zuazo et al. 2019. EMBO Mol Med) in which all the criteria for patient classification regarding baseline Thd was introduced. We have erroneously used as a comparative flow cytometry graph (right flow cytometry graphs) the profile of a G1 patient. We should have exemplified this with a non-responder G2 patient. The changes in the manuscript and the inclusion of an additional figure will improve the manuscript and make it understandable. This was also a problem with the other reviewers. Therefore, to clarify all these issues, we have introduced the following changes:
We have introduced essential information in the introduction regarding the classification of patients into G1 and G2. These patients can be readily identified at baseline by the proportion of CD4 Thd cells. G1 patients have >40% of CD4 Thd cells and G2 patients have <40%. To clarify this point we have added in line 80 “Indeed, G1 and G2 patients can be readily identified by quantifying the relative percentage of CD4 THD cells within CD4 cells in peripheral blood before starting immunotherapies. While G1 patients are characterized by having more than 40% of CD4 THD, G2 patients show less than 40% CD4 THD (REF PAPER ZUAZO). This classification constituted a first criterium for identification of non-responder versus potential responder patients.” In line 103 we have added the following: “presented a G1 profile (>40% baseline CD4 THD) and 31 patients (44.3%) had a G2 profile (<40% baseline CD4 THD) (Table 1).” The association of baseline CD4 THD with responders, non-responders and hyperprogressors therefore needed an additional figure (Figure 2 in the new manuscript) to clarify that CD4 G2 profiles (<40% baseline CD4 THD cells) are first associated to hyperprogressive disease, but not sufficient to differentiate progressors from hyperprogressors. This new figure also includes the statistics table originally present in figure 1C.The corresponding text has been included, that will clarify all sufficient data to fully understand our paper without having to read our previous publication. Hence, we have added in line 114 the following “The association of HPD with G1/G2 CD4 T cell profiles was analyzed in the whole cohort of patients (Figure 2A). While objective responders were found with G1 patients (>40% baseline CD4 THD), HPD was detected within G2 patients. No significant association was observed with baseline CD8 THD profiles (Figure 2B). Indeed, HPD very significantly associated with a baseline G2-type of systemic CD4 T cell profile as defined by Zuazo et al (p = 0.003 by Pearson's Chi Square test) within progressors, which identifies patients with dysfunctional CD4 immunity before the start of immunotherapy [8].”
To explain the link from our initial observations to analysis of changes from baseline to first cycle of therapy, we added in line 136 the following: “Although a significant correlation of HPD with a baseline G2 profile was observed (Figure 2), it was not sufficient to separate G2 progressors from hyperprogressors. Others and we have shown that a systemic expansion of CD28+ CD4 T cells following the start of immunotherapy is characteristic in responder patients. Therefore, we decided to find out if changes in the highly differentiated CD28-negative (THD) CD4 populations from baseline to first cycle of therapy would be an indicator to further discriminate HPD from non-HPD progressors.”
The flow cytometry graphs and data are highly confusing, as well as the legend. This has been caused by two things: a mistake from our part as CD28+ CD27+ in the legend should have read CD28negative CD27 negative cells. Secondly, to erroneously using as a comparative example a G1 patient, which is not adequate. This is a clinically-oriented paper and most of the cytometry data was already shown in our previous publication. Therefore, we have corrected the flow data including flow cytometry from a G2 non-hyperprogressor without a CD4 THD burst. Figure 2 is now in the text Figure 3, and we have made sure that all parts of the figure are referenced in the text. The figure legend has been corrected accordingly, to make it clearer.
In Figure 2C, the lines indicating the comparisons between groups are confusing. If this study was approved by an Institutional Review Board (IRB), the approval number should be mentioned in the manuscript.
We have improved the figure (and the rest of the figures in the manuscript) to make it easier to follow, now Figure 3. We sincerely apologise for having forgotten to include the information on the approval, already present in our previous publication. Our study was approved by an institutional Review Board (detailed in our previous paper). We have added this in Materials and Methods as follows: line 290 “The study was approved by the Ethics Committee at the Hospital Complex of Navarre (Pyto2017/49). Informed consent was obtained from all subjects and all experiments conformed to the principles set out in the WMA Declaration of Helsinki and the Department of Health and Human Services Belmont Report. Samples were collected by the Blood and Tissue Bank of Navarre, Health Department of Navarre, Spain.”
Reviewer 2 Report
Reviewer comments on cancers-680067
In the manuscript by Arasanz et al., the authors report the diagnostic biomarker which predicts the hyper progressive diseases (HPD) in non-small cell lung cancer patients treated with anti-PD-1/PD-L1 immunotherapies. The authors performed flowcytometry of peripheral blood mononuclear cells obtained before the first and before the second cycle of treatment. They found that the proportion of post-treatment / pre-treatment CD28- CD4 T cells (highly differentiated CD4 T cell (T HD)) was significantly higher in HPD when compared with non-HPD patients. Moreover, an increase of CD28- CD4 T cells ≥ 1.3, which the author defined as CD4 THD burst, was significantly associated with HPD and this cut-off value identified HPD with 82% specificity and 70% sensitivity.
Because HPD can be severe disadvantage for patients and predictive biomarkers of HPD are desired, the focus of this manuscript is interesting and important. However, the number of HPD patients analyzed in this study is too small to discuss about the value of CD4 THD burst. Although there is a statistical significance regarding to a cut-off value of ΔCD4 THD ≥ 1.3, the difference between HPD and non-HPD seems modest. In figure 2A, ΔCD4 THD shows higher in HPD (left) than non-HPD (right), however, the percentage of baseline CD4 THD is much higher in non-HPD as compared to HPD. If CD4 THD plays an important role in the onset of HPD, this data is confusing.
Author Response
RESPONSES TO REVIEWER 2
We appreciate the comments from Reviewer 2, who considers that the focus of our manuscript is interesting and important. The reviewer 2 raises three reasonable concerns. One of them has been addressed for Reviewer 1 and 3, and was caused by an error in the previous Figure 2, and the assumption that a reader of this paper would also have read our previous recent paper (Zuazo et al. 2019. EMBO Mol Med). As such, the manuscript needed to incorporate some data to clarify it. We have addressed the Reviewer´s concerns as follows:
The number of HPD patients analzyed in this study is too small to discuss about the value of CD4 THD burst.
Fortunately, HPD is not a frequent adverse event, although there is some discrepancy on the actual percentage depending on the cancer type and study. We would like to point to the attention of the Reviewer that although the number of HPD is limited, but this is the first prospective study carried out with a dynamic systemic biomarker. This fact adds value to the data.
Nevertheless, it is not our intention to “over-conclude” on the data. We need to stress to the readers that this is an exploratory study, and as such, the results are not diagnostic but hypothesis-generating and should be taken objectively with caution (see change in title). Therefore, to clarify these issues to the reader and for the sake of objectivity, we have explained the context of our study and discuss on the number of HPD patients as follows:
We have changed the title of the paper, as explained in the response to Reviewer 1. Line 237: “Therefore, in this exploratory study we decided to monitor if an increase in CD28-negative populations” Line 273: “Nevertheless, our prospective study is exploratory in nature with a limited number of HPD patients. In addition, identification of HPD is challenging and not all studies follow the same criteria. Indeed, many HPD patients progress so rapidly that they cannot be objectively diagnosed before death. Therefore, if validated, quantification of this systemic immunological phenomenon would urge to interrupt treatment.”
Although there is a statistical significance regarding to a cut-off value of ΔCD4 THD≥ 1.3, the difference between HPD and non-HPD seems modest. In figure 2A, ΔCD4 THD shows higher in HPD (left) than non-HPD (right), however, the percentage of baseline CD4 THD is much higher in non-HPD as compared to HPD.
We apologise to the Reviewer. Indeed, as explained to Reviewer 1, the flow cytometry plots from the non-HPD were erroneous, and belong to the G1 cohort and not to the G2 cohort. That was our mistake. Therefore, we had to introduce more background information and additional data to clarify interpretation of this biomarker, without the necessity of reading our previous paper (Zuazo et al. 2019. EMBO Mol Med).
The context of the study is as follows: In our previous paper we defined G1 and G2 patients according to CD4 functionality and the baseline percentage of CD4 THD cells (See new Figure 2). G1 patients have >40% of baseline CD4 Thd cells and G2 patients have <40%. G1 patients are mainly responders. None of the G2 patients responded to therapy. Now, we have introduced another figure in which the percentage of baseline CD4 THD cells used to define G1 and G2 cohorts was plotted as a function of TGR and responses. This graph highlights the distribution of responders, non-responders and HPD as a function of G1 or G2 profiles, and TGR. HPD patients are located within the G2 cohort. However, a G2 profile was not sufficient to discriminate HPD from G2 non-responders. The reasoning for further analysis of changes from baseline to first-cycle resides in this data.
This data is now the new Figure 2. Basically, HPD patients are found within G2 patients (New Figure 2). This was mentioned in the text, but we believe that the introduction of this figure will clarify this to the reader and the Reviewer. However, a G2 profile was not enough to differentiate HPD patients from G2 progressors. Therefore, we decided to find out if changes from baseline to first cycle of therapy would be an indicator. To clarify this rationale we have introduced the following: Page 136: “Although a significant correlation of HPD with a baseline G2 profile was observed (Figure 2), it was not sufficient to separate G2 progressors from hyperprogressors. Others and we have shown that a systemic expansion of CD28+ CD4 T cells following the start of immunotherapy is characteristic in responder patients. Therefore, we decided to find out if changes in the highly differentiated CD28-negative (THD) CD4 populations from baseline to first cycle of therapy would be an indicator to further discriminate HPD from non-HPD progressors. “ Hence, the proper context for comparing Thd bursts would be G2 patients, and we erroneously added a G1 patient in Figure 2A, right plots. We have corrected the data and introduced (now Figure 3A and 3B) the proper comparative control, a G2 non-HPD progressor without a CD4 THD burst. The Figure legend has been also improved to make it more understandable. If CD4 THD plays an important role in the onset of HPD, this data is confusing.Our intention was not to claim that CD4 THD play a role in the onset. It is a dynamic biomarker that increases within G2 patients in HPD, which may or may not be implicated in the onset.
Therefore, to clarify this point we have added the following sentence in discussion:
Line 285, Discussion: “Although our study does not address whether CD4 THD cells are directly involved in the onset of HPD, these cells provide a biomarker that changes within the G2 cohort in HPD patients.”
Reviewer 3 Report
In this manuscript, Arasanz et. al. described changes to CD4 T cell populations between cycles 1 and 2 of PD-1 checkpoint inhibition in NSCLC patients. Changes in the T cell landscape have been previously reported in lung cancers following ICI therapy; however, it is interesting that significant changes between HPD and non-responders were present, but not between non-responders and responders. This study may be useful to clinicians in the future as a potential biomarker for identifying patients susceptible to HPD. I do have some reservation regarding the study as identified below:
Overall, I would have liked to have seen further staining for T cell markers such as CD8 CTLs or even CD4-positive subsets (Th1, Th2, Th17, etc.) to provide more evidence of the changing lymphocyte landscape during progression vs response/non-response with ICI. This is particularly important as increases in some CD4 subsets have been associated with improved clinical outcomes (i.e. Th1) whereas others (Tregs) may suggest a poor prognosis.
As suggested in line 51, there are other cell types other than T cells that play a significant role in immunosuppression/disease progression. The study does not provide any clear evidence that CD4-positive T cells are driving the observations of disease progression in this study. In this regard, the study appears relatively weak as it does not account for alterations in other APCs, NK cells, or alternative T cell phenotypes that interact with CD4 T cells to mount an immune response. This information would significantly strengthen the manuscript.
Table 1 should also present patients in which information is “unknown” for clarity. Not all categorical variables add up to 70 patients (see EGFR status), thus, it would be beneficial to see how many patients do not have data available.
It is worth noting that the system for classifying CD4 cell functionality (G1/G2) as referenced in [8] was developed by the same lab in which the manuscript in review was submitted. Has this system been validated by outside sources?
Line 132 – This suggests that expansion of differentiated CD4 cells regulate HPD, but not response to ICI. This is interesting as CD4 T cells recognize neoantigens that can drive a response to immunotherapies. Is data regarding tumor mutation burden available for these patients? This may shed some light on why there is no difference in CD4 HD expansion among responders and non-responders.
Figure 2 – the samples taken from patient with HPD shows a significantly lower CD28- CD27- compartment, while the sample taken from the non-HPD patient shows the majority of cells stained negative for both markers. Is data available to show if there are significant differences in the baseline staining (prior to ICI) between HPD and non-HPD patients? Perhaps it would be beneficial to show the absolute counts for each T cell phenotype?
Line 167 - For statistical analysis, why was adjusting for age, race, and comorbidity scores not performed? A significant association between age and HPD has been previously described and warrants adjusting the survival estimates to reflect this potential confounding. The materials and methods section does not describe whether or not the survival analysis and Cox regression was adjusted for potential confounding variables. Additionally, stratified analysis based on histology, smoking status, gender, etc. is generally performed and can be added as supplemental data.
Figure 3 – I am not sure what Panel E and F provide to the manuscript as patients presenting with HPD would not be expected to exhibit improved OS or PFS when compared to non-responders not presenting with HPD. Additionally, I would suggest making the axis and text for Panel F the same size as the rest of the Panels on the Figure.
Is there a reason why tumor growth rate, but not the appearance/growth of new lesions was the criteria for HPD?
Author Response
RESPONSES TO REVIEWER 3
We thank the Reviewer for the positive comments. The Reviewer considers our work to be useful to clinicians in the future as a potential biomarker. The Reviewer has also some important concerns that we have addressed as follows:
Overall, I would have liked to have seen further staining for T cell markers such as CD8 CTLs or even CD4-positive subsets (Th1, TH2, Th17) to provide more evidence of the changing lymphocyte landscape during progression versus response to ICI. This is particularly important as increases in some CD4 subsets have been associated with improved clinical outcomes (Th1) whereas others (Tregs) may suggest a poor prognosis.
We assumed that a reader of the current manuscript would have read our previous paper (Zuazo et al. 2019. EMBO Mol Med) where we addressed many of these points. We are aware now that more background information on our previous paper needs to be introduced in our current manuscript. We did not include this information in the current manuscript as it has been published.
Briefly, we found that CD8 T cells and their changes were not biomarkers of response vs progression, here and in our previous paper. Nevertheless, to comply with the Reviewer suggestion, we have added a new figure (Figure 2) where we have included CD8 THD cell quantification vs clinical response and TGR. This data has not been published before and we hope that this will complete our current manuscript. Regarding CD4 subsets, we did an extensive characterization of Th profiles in the whole cohort of patients, published in Zuazo el al. 2019. Systemic CD4 T cells corresponded to Th17, in agreement with other studies, without differences between responders and progressors. We found that progressors showed CD4 T cells which were dysfunctional in proliferation but not in cytokine production, or even in specificity towards lung cancer antigens. Hence, to clarify this point we have added the following in the text, to clarify that these studies have already been performed and published: Line 84, introduction: “We also found that all patients presented systemic CD4 T cells with mainly Th17 phenotypes, without differences between responders and progressors, and without significant differences in the percentage of lung-cancer specific CD4 or CD8 T cells (REF ZUAZO). However, CD4 T cells from progressors were highly dysfunctional in proliferation but not in cytokine production.”As suggested in line 51, there are other cell types other than T cells that play a significant role in immunosuppression/disease progression. The study does not provide any clear evidence that CD4-positive T cells are driving the observations of disease progression in this study. In this regard, the study appears relatively weak as it does not account for alterations in other APCs, NK cells, or alternative phenotypes that interact with CD4 T cells to mount an immune response.
This is an exploratory, hypothesis-generating study of a biomarker which may or may not be driving HPD. Our intention was not to claim that CD4 THD play a role in the onset. It is a biomarker that increases within G2 patients in HPD, which may or may not be implicated in the onset. The main focus of this study was not to provide a mechanistic model for HPD onset, but a clinically-oriented study to help identification of this patient population. In addition, our study puts forward experimental evidence indicating that this CD4 T cell subset should be explored much further in HPD. Hence, the strength of this paper is that this is the first (as far as we are aware) prospective study with HPD for a dynamic T cell biomarker.
We did analyse changes in other cell populations, although we have not mentioned this in the paper or provided the data. The analysed populations were granulocytes, neutrophils and absolute T cell counts from clinical data for each patient. We did not (in our current cohort) observed any difference in HPD compared to other patient groups in the dynamics of these other populations. Hence, the data was not included as it was negative, but I agree we should have mentioned this at least in the discussion section.
Therefore, to clarify this point we have added the following sentences:
We have changed the title of the paper, as explained in the response to Reviewer 1. Line 237: “Therefore, in this exploratory study we decided to monitor if an increase in CD28-negative populations” Line 285: “Although our study does not address whether CD4 THD cells are directly involved in the onset of HPD, these cells provide a biomarker that changes within the G2 cohort in HPD patients. ” Line 266: “In this study we also quantified the relative proportions and changes of other systemic immune populations including neutrophils, monocytes, HLA DR+ cells and absolute T cell counts. No significant associations were found, as corroborated also by lack of association with neutrophil-to-lymphocyte ratio, a classical prognostic variable.
Table 1 should also present patients in which information is “unknown” for clarity. Not all categorical variables add up to 70 patients (see EGFR status), thus, it would beneficial to see how many patients do not have data available.
There was an error in the table, and now after replacing the table all categorical variables add up to 70 patients.
It is worth noting that the system for classifying CD4 cell functionality (G1/G2) as referenced in (8) was developed by the same lab in which the manuscript in review was submitted. Has this system been validated by outside sources?
We do appreciate the question and the interest. It is currently under validation. It is being expanded to other ICI treatments (first-line, chemo-ICI combinations, etc).
Line 132- This suggests that expansion of differentiated CD4 cells regulate HPD, but not response to ICI. This is interesting as CD4 T cells recognize neoantigens that can drive a response to immunotherapies. Is data regarding tumor mutation burden available for these patients? This may shed some light on why there is no difference in CD4 HD expansion among responders and non-responders.
We do thank the reviewer for this appreciation and suggestion. Indeed, that was a question that we addressed in our previous paper (Zuazo et al. 2019), but not by having data on tumor mutation burden (which unfortunately we do not have due to lack of sufficient biopsy material from many patients). We did quantify the percentage of lung cancer-specific CD4 and CD8 T cells in peripheral blood from responder and non-responder patients. There were no significant differences in the percentage of antigen-specific CD4 or CD8 T cells between responder and non-responders, or with Th phenotypes. The only difference that we found was on proliferative capacities.
We think that this is a good point raised by the Reviewer, and we have completed our manuscript by adding that antigen-specificity of CD4 and CD8 T cells did not discriminate between responders and non-responders. Therefore, we have added the following:
Introduction, line 84: “We also found that all patients presented systemic CD4 T cells with mainly Th17 phenotypes, without differences between responders and progressors, and without significant differences in the percentage of lung-cancer specific CD4 or CD8 T cells (). However, CD4 T cells from progressors were highly dysfunctional in proliferation but not in cytokine production”
Figure 2- The samples taken from patient with HPD shows a significantly lower CD28- CD27- compartment, while the sample taken from the non-HPD patient shows the majority of cells stained negative for both markers. Is data available to show if there are significant differences in the baseline (prior to ICI) between HPD and non-HPD patients? Perhaps it would be beneficial to show the absolute counts for each T cell phenotype?
As we did with the previous Reviewers, we sincerely apologise because the context of the study was not properly explained and the example provided was inadequate. We assumed that a reader of this paper would have read our previous one. Indeed, as explained to Reviewer 1, the flow cytometry plots from the non-HPD were introduced erroneously, as the patient on the right without HPD belongs to the G1 cohort and not to the G2 cohort. That was our mistake.
First, to clarify all these concerns, the context of the study is as follows: In our previous paper we defined G1 and G2 patients according to CD4 functionality and the baseline percentage of CD4 THD cells (See new Figure 2). G1 patients have >40% of baseline CD4 Thd cells and G2 patients have <40%. G1 patients are mainly responders. None of the G2 patients responded to therapy. Now, we have introduced another figure in which the percentage of baseline CD4 THD cells used to define G1 and G2 cohorts was plotted as a function of TGR and responses. This graph highlights the distribution of responders, non-responders and HPD as a function of G1 or G2 profiles, and TGR. HPD patients are located within the G2 cohort. However, a G2 profile was not sufficient to discriminate HPD from G2 non-responders. The reasoning for further analysis of changes from baseline to first-cycle resides in this data.
Therefore, we have plotted the % of baseline CD4 THD cells used to define G1 and G2 cohorts against TGR and clinical responses, and presented this data here in the new Figure 2.
This data is now the new Figure 2. Basically, HPD patients are found within G2 patients (New Figure 2). This was mentioned in the text, but we believe that the introduction of this figure will clarify this to the reader and the Reviewer. However, a G2 profile was not enough to differentiate HPD patients from G2 progressors. Therefore, we decided to find out if changes from baseline to first cycle of therapy would be an indicator. To clarify this rationale we have introduced the following: Page 136: “Although a significant correlation of HPD with a baseline G2 profile was observed (Figure 2), it was not sufficient to separate G2 progressors from hyperprogressors. Others and we have shown that a systemic expansion of CD28+ CD4 T cells following the start of immunotherapy is characteristic in responder patients. Therefore, we decided to find out if changes in the highly differentiated CD28-negative (THD) CD4 populations from baseline to first cycle of therapy would be an indicator to further discriminate HPD from non-HPD progressors. “ Hence, the proper context for comparing Thd bursts would be G2 patients in the old Figure 2, and we erroneously added a G1 patient in Figure 2A, right plots. We have corrected the data and introduced (now Figure 3A and 3B) the proper comparative control, a G2 non-HPD progressor without a CD4 THD burst. The Figure legend has been also improved to make it more understandable.
Line 167- For statistical analysis, why was adjusting for age, race, and comorbidity scores not performed? A significant association between age and HPD has been previously described and warrants adjusting the survival estimates to reflect this potential confounding. The materials and methods section does not describe whether or not the survival analysis and Cox regression was adjusted for potential confounding variables. Additionally, stratified analysis based on histology, smoking status, gender, etc is generally performed and can be added as supplemental data.
We had performed all suggested analyses, and results did not change our conclusions. We therefore addressed this concern as follows:
The cohort under study was Caucasian, so it was not necessary the adjustment for race. We have clarified this in the Study design as follows: Line 294: “Data and blood samples were prospectively collected from 70 Caucasian patients with locally advanced or metastatic NSCLC treated with ICI (nivolumab, pembrolizumab, atezolizumab) following current indications” As suggested by the Reviewer, we provide in the new manuscript all the information adjusted to potential confounding variables as supplementary figure, which includes stratification by age, ECOG, organ affectation, neutrophil-to-lymphocyte ratio, derived neutrophil-to-lymphocyte ratio serum albumin, serum LDH, Grim score and LIPI index. The reference to the new supplementary figure is found in line 114 as follows: “The association of HPD with G1/G2 CD4 T cell profiles was analyzed in the whole cohort of patients (Figure 2A) as well as with other variables (Supplementary Figure).”
Figure 3. I am not sure what panel E and F provide to the manuscript as patients presenting with HPD would not be expected to exhibit improved OS or PFS when compared to non-responders not presenting with HPD. Additionally, I would suggest making the axis and text for Panel F the same size as the rest of the panels on the figure.
We thank the reviewer for his/her opinion. Nevertheless, we feel that not many readers will find this to be so evident, so we decided to leave these panels. We have also improved this figure, and the rest of the figures in the manuscript to make them more readable.
Is there a reason why tumor growth rate, but not the appearance/growth of new lesions was the criteria for HPD?
We have used the most frequent criteria for HPD based on acceleration of growth rate, which is more conservative. For that existing lesions are needed to estimate growth rates.
Round 2
Reviewer 2 Report
The manuscript has been revised well. I think this manuscript will be acceptable.
Reviewer 3 Report
The authors, for the most part, have revised the manuscript to be suitable for publication in Cancers. While the study is largely exploratory and conclusions based on these data need to be validated in a larger cohort, it does lend some new evidence to induction of HPD in non-responders to ICI. For these reasons, the reviewer recommends that the manuscript is acceptable in its present form.